# A Healthy Diet Is Not More Expensive than Less Healthy Options: Cost-Analysis of Different Dietary Patterns in Mexican Children and Adolescents

**DOI:** 10.3390/nu13113871

**Published:** 2021-10-29

**Authors:** Patricia Clark, Carlos F. Mendoza-Gutiérrez, Diana Montiel-Ojeda, Edgar Denova-Gutiérrez, Desirée López-González, Laura Moreno-Altamirano, Alfonso Reyes

**Affiliations:** 1Clinical Epidemiology Research Unit, Children’s Hospital of Mexico Federico Gomez-Faculty of Medicine, National Autonomous University of Mexico UNAM, Mexico City 06720, Mexico; osteoclark@gmail.com (P.C.); nutriologadianamont@gmail.com (D.M.-O.); dradesireelopez@gmail.com (D.L.-G.); 2Nutrition and Health Research Center, Public Health National Institute, Cuernavaca 62000, Mexico; edenovag@gmail.com; 3Department of Public Health, Faculty of Medicine, National Autonomous University of Mexico UNAM, Mexico City 04510, Mexico; lamorealmx@yahoo.com.mx; 4Center for Economic and Social Studies in Health, Children’s Hospital of Mexico Federico Gomez, Mexico City 06720, Mexico; ceeses.himfg@gmail.com

**Keywords:** cost-analysis, dietary patterns, cost-benefit analysis, economic evaluation, pediatric, children, adolescents

## Abstract

Unhealthy diets are recognized as a major risk factor for many diseases. The decrease in costs of industrialized products, as well as the possible misinformation about a healthy diet, has led to new behaviors in the dietary patterns of the pediatric population. The costs of dietary patterns have not been estimated in our population, so the objective of this study was to determine the cost associated with dietary patterns in Mexican children and adolescents, hypothesizing that a healthy diet is not necessarily more economically expensive. This study analyzed data from a population-based cross-sectional study of healthy children and adolescents in Mexico City. Data were collected from a food frequency questionnaire and the meal cost of habitual food shopping. Eating patterns were obtained by using principal component analysis. A micro-costing technique was performed to obtain the direct costs of each pattern. When comparing the healthy pattern with the transition and non-healthy patterns, it was observed that there were no statistically significant differences between the dietary patterns (*p* = 0.8293). The cost of the healthy pattern only takes up 16.6% of the total biweekly income of a salaried Mexican. In this study, no differences were observed between the costs of a healthy and a less healthy diet.

## 1. Introduction

Non-healthy diets are widely recognized as a determinant risk factor in the development of chronic, noncommunicable diseases (NCDs) such as obesity, diabetes, type 2 mellitus, cardiovascular disease, and many types of cancer that greatly contribute to the global load of NCDs in different populations [1,2]. In Mexico, obesity is one of the biggest health problems in our pediatric population where 35.6% of children and 38.4% of adolescents are overweight or obese [3], with Mexico being the country with the highest prevalence globally of children with this health condition [4].

Eating behavior can be described by dietary patterns; this methodology has had an important boom since the past decade, because through this type of analysis, it has been possible to define the dietary characteristics of populations, describing consumed food groups, their included nutrients, their combination and variety, and the frequency and quantity of food consumption [5]. For this reason, the use of dietary patterns has taken relevance in the field of public health since these allow for more accurate recommendations about the population’s nourishment [6,7].

Socio-economic and cultural environments are the main factors that have affected Mexico’s food production, which has resulted in a direct modification of dietary patterns. In terms of production systems, Mexico underwent a modification in food commercialization models in the last decades, leading it from being a self-sustainable country in food-related issues to one that depends upon importation to feed its population. One of the leading consequences of these events has been a rise in food cost [8].

Due to these modifications in the systems of production, an interest in knowing how prices influence Mexican consumers’ food choices has recently incremented. In some cases, the existence of a perception about healthy diet is noted, in which it is believed that certain foods are healthier, more expensive, and therefore less accessible. Such a belief has been a constant obstacle in the promotion of healthy diets [1,9,10,11].

On the other hand, the cheapening of energy-dense foods is associated with the use of additives and, in many cases, is also due to the use of low-quality ingredients. Additionally, the incremented use of sophisticated and focalized marketing has been seen, along with a consequent increment in the ease of access to ultra-processed foods, even in rural areas far removed from cities. The combination of all this led to an increase in the demand for these products, and, consequently, costs decreased significantly [12,13,14], resulting in a gradual change in the dietary patterns in the Mexican population, especially in the dietary behavior of the children and adolescent population [15,16,17].

In this sense, the literature reports inconclusive data about the association between cost, diet content, and nutritional quality, as it has been noted that in many cases, healthy diets can be more expensive [18,19,20,21,22]. Nonetheless, this controversy is refuted because the conclusions reached in these studies differ in their methodologies and metrics, where the comparison of results to generate consistent findings is barely made possible. In the published literature, there are more than a few methodologies to estimate the cost of diets. The most common method is based on a rate between food price and the calories to calculate the cost per calories. Another less common method estimates the price per portion of food by dividing the cost per package between the number of pieces. Furthermore, most studies published are a secondary level analysis based on official surveys and public price databases [6,11,23,24,25,26,27,28,29,30]. Additionally, the costs of a healthy diet appear to be higher because the components of the less healthy diets are based on poor quality products or contain, for example, corn and soybean as their main ingredients, which indeed lower the cost of production. Usually, the diets with lower costs are as well the ones with fewer nutrients and a higher calorie content [9,21].

Several studies are published in our country regarding the dietary patterns in adults and their relationship with some diseases such as gastric cancer, cardiovascular disease, and obesity [31,32,33,34,35,36]; nonetheless, none of these studies included any economic analysis; for the population of children and adolescents, there are some studies that report dietary patterns [37,38], but none of them reported their costs. The only study that reported the costs of diets in Mexicans was a study based on data from the 2012 ENSANUT survey [39] where authors explored the energy density of foods and their cost on adults, concluding that an average diet of 1958 kcal/d had a cost of MXN 49.00 (USD 2.78) per day [40].

In a recent study from Mexico [41], prices of food in seven consecutive years were reported (2011–2018) where no increment in costs was observed in healthy foods, and some increments were found in non-healthy choices; however, the main conclusion of this study was that no significant price changes over this time were observed. Therefore, it is unknown if the cost, as it has been mentioned in other studies [7,31,32,33,34,35,36,37,38], is the variable that prevents the purchase of healthy food for Mexican families, or other factors of a different nature besides the costs are the barriers that prevent the selection of healthy dietary choices. Moreover, it is well known that the consumption of food and nutrients differs between adults, children, and adolescents because of biological, social, and cultural aspects, and these elements must be considered for this specific population [42,43].

Based on the above considerations, and due to our country’s lack of information in this field, as well as the epidemics of obesity in our population, the present study aims to determine the cost associated with dietary patterns in Mexican children and adolescents, posing the hypothesis that a healthy diet is not necessarily a more economically expensive one.

## 2. Materials and Methods

The present study used data from a population-based cross-sectional study aimed to determine the reference values of Mexican children and adolescents’ body composition. The sample was selected using information from the Secretariat of Public Education (SEP, for its acronym in Spanish) in Mexico City and the metropolitan area; a random sample of private and public schools was taken from the designated, geographical areas. Invitation letters were subsequently directed to each of these schools’ families’ parents. The interested participants booked an appointment to visit the Clinical Epidemiology Research Unit at the Federico Gomez Children’s Hospital (HIMFG) —Faculty of Medicine from the National Autonomous University of Mexico (UNAM), where it was verified through a clinical history that the participants were healthy and met the criteria to be included in the study. The inclusion criteria were healthy individuals, without known chronic, endocrine, systemic, respiratory, neurological, cardiac, or psychiatric disorders; individuals without chromosomal diseases, genopathies, dysmorphic syndromes; and individuals who were not receiving pharmacological treatment affecting their lipid or glucose metabolism. The detailed design of the methodology can be consulted elsewhere [44].

In addition to the measurements applied to obtain the body composition’s reference values, a questionnaire of food frequency (FFQ) about nourishment that was previously validated in the Mexican population [45] was given to children and adolescents, as well as questionnaires about the cost of foods and the food establishments where they supply, which were given to the families’ parents or guardians; this is detailed in the next paragraphs.

From a sample of 2104 children and adolescents aged 4.5 to 20 years, recruited for the study, “Reference values for bone mineral density in healthy Mexican children and adolescents” [44], complete information was obtained about the diet of 1955 children and adolescents (990 males and 895 females). Three diet patterns were derived with this information using the analysis of principal components; these patterns were defined as: (a) Healthy pattern, which consists of a high consumption of vegetables and fruits, and a sufficient consumption of non-industrialized cereals and protein of animal origin; (b) Transition pattern, which combines a moderated consumption of vegetables, fruits, non-industrialized cereals, protein of animal origin, and in addition, the regular consumption of refined sugar and saturated fats; and (c) Non-healthy pattern, in which the consumption of refined sugar, sugary drinks, and saturated fats dominates.

For the obtainment of costs, a descriptive analysis of the direct costs of the 146 foods in the diet questionnaire through the micro-costing technique was carried out gathering the price and calculating the cost for each product as mentioned in Figure 1. The individual costs of the foods were later transformed into servings and equivalents to determine the total cost of each one of the three patterns.

The anthropometric measurements and the application of the instruments and questionnaires were applied by a multidisciplinary group conformed by pediatricians, dietitians, and health economists. This study was carried out complying with the Declaration of Helsinki in clinical research on humans [46] and was supervised and approved by the Investigation, Ethics, and Biosecurity Committees at HIMFG (HIM2015-055). The parents or legal guardians of the children/adolescents, as well as children over 7 years old, signed the informed consent to participate.

### 2.1. Measurement Instruments

Food frequency questionnaire (FFQ): The FFQ used for this study was previously validated in the Mexican population [45]. It consists of 116 items, divided in subgroups according to the type of food they belong to: vegetables, fruits, cereals, tubercles, legumes, products of animal origin, oils and fats, sugary drinks, candies, and snacks. An average was taken for each food, specifying the portion’s size, slice, measuring glass/cup, or natural unit. The frequency of consumption used in the FFQ consists of the following answers: never, less than once per month, 1–3 per month, once a week, 2–4 times a week, 5–6 per week, 1 per day, 2–3 per day, 4–5 per day, 6 or more per day. The FFQ is based on a reminder of average food consumption 12 months prior to the day of its application [45]. For this study’s purposes, 30 foods targeted toward the children/adolescent population and easily available to them for quotidian consumption (candies and snacks) were added to the instrument, leaving a total of 146 items.Questionnaires dealing with socio-economic aspects, eating habits, and food stores/markets: a 20-item questionnaire was developed specifically for this study to get information regarding the social and economic aspects of the surveyed families. The questionnaire included information about the profession and occupation of the family parents or tutors, monthly monetary income, household characteristics, mode of transport taken, and amount of time dedicated to buying and preparing the foods. A second questionnaire asked about the different stores/markets where the families habitually get their food supply.

### 2.2. Dietary Patterns

The energy consumed from each food was converted into a portion of total energy consumption per day, and was later standardized using the score Z. The foods and drinks in the questionnaire were classified into 29 food groups, which were used as a base for the derivation of dietary patterns. In summary, the criteria used to assign a food to a particular food group were their similitude of nutrient content (e.g., fats, proteins, carbohydrates) and dietary fiber. Other groups were classified according to their fatty acids profile. (e.g., vegetable oils). Finally, some individual foods were groups of themselves because of their frequency of consumption and/or unique nutritional composition (e.g., Mexican foods and corn tortillas, potatoes, eggs, tomato juice, etc. [6,33,47]; Appendix A).

The dietary patterns were derived from the analysis of the food groups’ principal components. The resulting factors were rotated orthogonally (varimax rotation) for better interpretation. After evaluation of the eigenvalues, the scree plot test, and interpretability, all values >1.5 were retained. In addition, each factor was defined by a subset of at least five food groups with an absolute factor load ≥0.2 (considering that the absolute factor load ≥0.2 contributed significantly to the dietary pattern), as was suggested in prior analyses. The factor scores for each dietary pattern were calculated by adding the consumption of the different food groups weighted by their loading factor, and each participant received a score for each of the 3 dietary patterns. Once the three principal components were obtained, the patterns were disaggregated according to the quantity of daily consumed portions for their subsequent translation to costs. The methodological and procedural details of the statistical analysis were previously published by Denova-Gutierrez et.al [6,33,47].

### 2.3. Cost Analysis

To determine the economic impact of the children and adolescent population’s dietary patterns, a three-step methodological strategy was carried out using the sequence that can be observed in Figure 1.

Determination of unitary costs: The costs were obtained in triplicate (acquisition costs) of the 146 foods collected in the FFQ in 3 different food stores/markets (supermarket, market, and convenience store) to know the variability that can be attributed to different brands and presentations (leading brand and/or private or distributor brands), using a fixed effects model to estimate the residual error. It should be noted that for this economic analysis, all the prices were obtained using a standard unit of 1 Liter or 1 Kilogram (depending on the product); additionally, it was assumed that the foods’ preparation is an established capacity.

$$\mathrm{Unitary}\;\mathrm{cost}=\frac{\sum^{\;}(C_{1,2,3}*W_{1,2,3})}{n}$$
where:

*C1…* = Cost of product i in the type of store/market 1.

*W1…* = Weight of product i (1/standard deviation).

2.Cost of consumed portion: In accordance with the proposed methodology, the unitary costs per each food presentation were transformed into portions using the Mexican Equivalents System [48] using the conversion factors technique, as shown below:

*Cost of the food’s unit portion = CUPi × FCi × Rei*
where:

*CUPi* = Unit cost of the product (food i) in its standard presentation.

*FCi* = Conversion factor for the food i.

E.g., (1 L/1000 mL) o (1 Kg/1000 mg) according to the type of food.

*REi* = Portion according to the Equivalent Food System.

As an example, a portion of milk’s cost calculation will be:
$$\mathrm{Cost of portion (milk)}\;=\;＄20\;*\left(\frac{1L}{1000}\right)*240=4.8\;\mathrm{Mexican}\;\mathrm{pesos}/\mathrm{portion}$$

Moreover, once the costs per portion for each food were established, they were multiplied by each food’s number of consumed portions within each of the three patterns using the following equation:$$\mathrm{Total}\mathrm{cost}\mathrm{of}\mathrm{pattern}X\;\;=\sum^{\;}\left(FCP_{ijk\ldots }*CRU_{ijk\ldots }\right)$$
where:

*FCPi* = mean frequency of product i’s consumed potions within pattern x

*CRUi* = Cost of unitary portion

3.Determination of the dietary patterns’ total cost: In this third phase, an arithmetic summation of the costs that can be attributed to each food was carried out to conform to the dietary pattern, as is mentioned in Figure 1. In addition, the confidence intervals with distribution Z are presented as shown in the following equation:

(x¯−zα2σn,x¯+zα2σn)
where:

$\bar{x}$ = Sample mean.

zα2 = Critical value.

$\frac{\sigma }{\sqrt{n}}$ = Standard error.

### 2.4. Statistical Analysis

Conventional descriptive statistic methods were used to report the sample’s demographic characteristics. Mean values with standard deviations are presented, or in absolute numbers with percentages according to the nature of the variables. The principal component analysis technique was used for the dietary patterns [6]. Simple mathematical operations derived from the formulas described earlier to define the unitary costs, cost per portion, and the summation of the costs of each of the three patterns were used to determine the costs of the diet. This study’s final unit of analysis was the cost of the dietary pattern.

## 3. Results

The socio-demographic characteristics of the sample in the present study are detailed in Table 1. It was observed that weight and size are greater in boys than in girls; nonetheless, corporal fat proportion was greater in girls. Men reported greater consumption of total calories per day.

### Costs of Diet

The recruitment period of the sample was from March 2015 to April 2019, and prices were collected between January 2019 and April 2019. Different pricing for the same product according to brand, area, and packing size was identified, as was noted in the methods. A total of 133 unitary costs according to the foods used for this study in the FFQ were estimated (the price of alcohol and a few foods and products that are found twice in the questionnaire were not costed).

Table 2 shows an example of 12 foods, along with the measurement unit, the unitary costs of each product, the conversion to a portion, and its equivalent are observed. For example, 1 average liter of milk has a unitary cost of MXN 20.05; the portion equivalent to a glass contains 240 mL, and its equivalent conversion is that of 0.24. The unit costs of the 133 foods in the FFQ can be found in the Appendix A.

Table 3 shows the 29 food groups incorporated into the three consumption patterns. The mean and standard deviation in portions for each food group and for each pattern are presented. Important differences can be observed; for example, the consumption of fresh fruits was 14.4 portions in the healthy pattern, which is considerably greater than the other two patterns’ average consumption, 9 in the transition pattern, and 9.4 in the non-healthy pattern.

The results of the average cost analysis for each of the three consumption patterns derived from the unitary cost of each food are observed in Table 4. The healthy pattern had an average cost of MXN 352.69, while the transition pattern had an average cost of MXN 323.65 and the non-healthy pattern MXN 311.43; there was no statistically significative difference observed between the dietary patterns (*p* = 0.8293). Its equivalent in American dollars and the minimum and maximum ranges are shown in Table 4 and in Figure 2. Considering that a Mexican person’s biweekly, minimum salary, as established by the CONASAMI [49], is that of MXN 2125.50, the cost of the healthy pattern takes up 16.6% of the total, biweekly income of a salaried Mexican. The percentages of the transition and non-healthy patterns’ spending were 15.2% and 14.7%, respectively.

## 4. Discussion

This study demonstrated that the average cost of consumption dietary patterns is not statistically significant; it was found that the difference between the most expensive and the least expensive dietary pattern was that of MXN 41.26 (USD 1.95). As far as we are aware, this is the first study that evaluates the costs and dietary patterns in Mexican children and adolescents.

International and national literature on this topic shows widespread heterogeneity regarding the methodology used to determine the costs related to diet, which is why the comparisons are not all that feasible. Most of these studies correspond to the following three approximations:

1. Calorie cost evaluation: This approximation contemplates energy (Kcal/kJ) as a unit of measurement, which allows comparisons to be made with other recommendations that standardized 2000 kcal as a mean daily consumption. Most studies are based on models of 100 g/100 kcal, or on the percentage of energy provided by the macronutrients [50]. In regular terms, the healthiest diets are calorically less dense, while diets with a high quantity of energy show a lower quantity of nutrients (vitamins, minerals, and fiber) and a high quantity of saturated fats and sugars [51]. For example, vegetables and fruits are foods that provide a lower quantity of calories per unit cost of food, being healthier, while ultra-processed foods—such as pastries, snacks, canned soup, sausages, sandwich bread, etc.—which have a greater quantity of energy per portion and also have a lower unit cost but contribute to a higher number of calories and a lower quantity of nutrients [11,24,52,53]. This methodology was conducted by Mendoza et. al. in Mexico in which results of the Health and Nutrition National Survey of 2012 (ENSANUT) [39] were analyzed. In this study, The National Institute of Statistics and Geography of Mexico National Index of Consumer Prices was used to obtain the prices of foods, and prices were assigned for every 100 g of the edible portion to each food. The monetary cost of the diets at an individual scale was calculated by multiplying the weight of each food by its unitary cost, and later, all foods and drinks consumed by each person were added; in addition, a total of 1800 kcal was used as standardized energy. The conclusion of this study was, according to this methodology, that diets that are richer in energy cost less, while diets with a lower energetic density cost more [40]. Some investigations established that the usage of Kcal to determine the cost of a diet could be an equivocal methodology seeing that the relation between energy density and the cost of foods depends on the food category [10].

2. The method of food substitution: It consists of comparing consumed foods from information collected using tools such as the 24 h reminder or the FFQ. This method generated scenarios of modified diets where a “standard” food is substituted by a “healthy” one, e.g., white bread versus low fat and sugar whole-grain bread, supplemented with vitamins and minerals, etc. In this sense, it is seen that products that are labeled as healthy tend to have a higher cost. The cost of foods for this type of analysis is obtained directly from main supermarkets, or indirectly from food databases in each locality. Through this methodology, the costs of “modified diets” are established by carrying them out. These were conducted in various countries, such as those in the United Kingdom and Australia. The results show that, depending on the composition of the diet, it can be more expensive if it is healthy; nonetheless, it also shows that better structured diets can be healthy at a lower cost [25,27].

3. Costing by food groups or dietary patterns: This method, as was previously mentioned, consists of grouping foods according to their nutritional characteristics, for example: fruits and vegetables, animal protein, fats, etc. With relation to the dietary patterns, this methodology has the advantage of reflecting a population’s diet diversity, which is why a better correlation between the diet and the foods’ cost can be observed, showing a more complete picture of what the population’s consumption tendencies are.

The present study was conducted following the third approximation, using the derivation of dietary patterns. There were only two similar studies found to this one in the literature [54,55], where the comparison can seem reasonable. The first study was conducted in the United Kingdom in a sample of 35,000 women from a cohort [54]. This work implemented a FFQ with 217 items that reported the consumption of foods 12 months prior to its application. The foods’ costs were taken from “The Diet and Nutrition Tool” (DANTE) database [56]. Seven dietary patterns were identified and evaluated using the Eatwell Plate, established by the UK’s health system, with the objective of analyzing the percentage of attachment to portions and food groups in this plate in each one of the patterns. A significant association between quality and cost in the diet was observable (*p* ≤ 0.001). The “Health Conscious” pattern was found among the seven analyzed patterns, which consisted of bran foods, potatoes, whole-grain foods, yogurt, low-fat dairy products, legumes, fish, vegetables, salads, and fruits; it showed twice the acquisition cost to that of the “Monotonous Low-Quality Diet”, which consisted principally of white bread, milk, and sugar. The range of the diet cost in these diets was £3.29 to £6.63/day. Since this study considered the analysis of different dietary patterns, if the results are not adequately interpreted, they could appear biased; for example, the “Health-conscious” diet exceeded the recommendations for a healthy diet of the Eatwell Plate by 66.6% and had a cost of £ 6.63 ± 1.95/day. A pattern that did achieve 100% with the recommendations of the Eatwell Plate was that of the “Conservative Omnivore”, which included almost every food, but mainly potatoes, meat, fish, eggs, fruits, and vegetables. This pattern had a cost of £4.14 (±£1.02).

Therefore, considering the “Conservative Omnivore” as the healthy pattern, it can be observed that there are no cost differences between this pattern and the “Monotonous Low-Quality Diet” pattern, seeing that the costs of these patterns are superimposed (£4.14 ± £1.02 vs. £3.24 ± 0.95). According to this study, it is demonstrated that following a healthy diet in accordance with the recommendations of the Eatwell Plate, a similar or equal cost to that of a non-healthy diet is possible. It is not possible to analyze the correlation between foods and the patterns’ derived portions in Mexico because the Mexican “plate of Good Eating” only represents a qualitative recommendation [57].

The second study was carried out in Canadian children [55]. Bukambu E. et al. analyzed the association between the cost and diet quality of Lower School children in a sample of 2731 students. The recollection of consumed foods was conducted using a FFQ (Harvard Youth Adolescent Food Frequency Questionnaire) [58]. With the information derived from the FFQ, the diets were classified accordingly to their quality in low, moderated, and high. Information about the foods’ costs was obtained by visiting four main supermarkets in Alberta, Canada. It was observed that the healthier the diet, the more expensive it is. The difference in costs between the higher quality pattern versus the lower quality one was 1.39 Canadian dollars per day. The authors report that in the long run, this difference would be of approximately 83.00 dollars per month, and 1014.00 dollars per year; for this reason, they consider that this is a factor that can be a barrier for economically disadvantaged groups. Nevertheless, it is important to mention that even though these results’ statistical analysis seem to show a significant difference, the minimum and maximum costs overlap, so that this difference could or could not exist between the three diet quality groups considered in this study.

The present study utilized dietary patterns and cost by portions, and with this, it was possible to demonstrate that a healthy diet in our children and adolescent population is not more expensive than a non-healthy diet, and, as discussed above, there are methodological reasons why other approaches and analyses may be less suitable for determining the cost of a healthy diet. As was mentioned in this study, we think that the use of dietary patterns has advantages over the other methodologies since it reflects the usual daily diet and represents in a way closer to the reality of the habits of eating and buying food of the Mexican families.

The methodology and results of other studies performing a calorie cost evaluation [11,19,20,22,51,52,53] should be taken with caution, as mentioned by Davis G. and Carlson A. [59], especially when analyzing the dietary components of populations whose diets are based on processed or ultra-rich products, and foods high in simple carbohydrates and saturated fats, as with most of the Mexican population [60]. This situation is emphasized by Ortiz-Hernández [61], who reports that for several years, this high-calorie, low-nutrient dietary pattern has been more economical due to the inherent characteristics of its components (e.g., refined carbs, simple sugars, and saturated fats) entailing low-cost production, storage, and distribution.

Despite this scenario, a recent Mexican study by Batis C. et al. [62], with the objective of comparing the cost of the biweekly dietary baskets created with the tool DIETCOST for the Mexican population, reported that a diet with higher amounts of fruits, vegetables, legumes, and nuts and lower amounts of animal protein sources, sugar sweetened beverages, and discretionary foods is affordable as part of a basic Mexican food basket and does not exceed the costs of an unhealthy diet, as reported in the international literature [9]. These new findings are consistent with our results. Likewise, the study of Mendoza et.al. [40], which analyzed the results of the 2012 ENSANUT survey [39], determined that the monetary cost of energy-adjusted diets was MXN 36.00/day for a low socioeconomic level and MXN 53.51/day for the highest socioeconomic level (MXN 252 to 374.57 per week), so these data are consistent with our results where we found that the healthy pattern had an average cost of MXN 352.69.

Analyzing our results, it is relevant to consider that the age of the participants could have some impact on the few differences found between healthy and unhealthy diets. In this population, most of the decisions about food purchases and consumption are made by the parents or primary caregivers of the children; however, such decisions mainly affect the children, since as they enter adolescence there begins to be more freedom over the purchase and consumption of food [63].

Another possible reason why no differences were observed between food costs and dietary patterns may be due to the few variations in food costs in recent years, as reported by Batis C. et al. [41] where they analyzed cost trends in Mexican food from 2011 to 2018; in this study, they observed that the prices of unhealthy foods have increased slightly more than healthy ones; however, it is still not understood how these changes have impacted diet quality. Additionally, it is important to consider that the three dietary patterns found are not different in the food content. The main difference between them was the frequency of the consumption for each food.

To combat obesity and NCDs, the Mexican Government has taken some actions, such as modifying the nutritional labeling of processed and ultra-processed products, as well as increasing the taxes on sugar-sweetened beverages and nonessential energy-dense food. From this last action, it was observed that one year after the implementation of the taxes, the consumption of these products decreased by up to 5% [64]; however, more time is needed to know the true impact of these policy actions and how they will impact food expenditures on Mexican families.

Among the strengths of this study is the usage of dietary patterns in the examined children and adolescent population, which are closer to the real consumption of foods, which also facilitated their transformation into food portions. Additionally, the methodology used in this study to obtain the costs (costing by triplicate) allowed the cost of the foods to be closer to the reality and commonness of our population’s everyday shopping, which also reflected the purchase value of the purchased foods at different socio-economic levels.

As to the information collected for this study, both the FFQ as well as the 24 h reminders were obtained from the targeted population—that is, Mexican children and adolescents—which is why no assumption of the information was required as was the case in other studies [37,40]. It is important to consider the existence of social and cultural factors, in addition to costs, which may influence food selection and diet behavior, such as: the availability of the foods, palatability, beliefs and customs, education, and the supply or promotion of unhealthy foods and drinks at the time of acquisition and preparation, which should be considered when interpreting the dietary patterns.

There are also limitations acknowledged in this study. One of them is that the sample was taken exclusively in Mexico City and the Metropolitan Area, which is why it does not reflect the diet characteristics of rural areas or different regions within the country, which, as is well-known, may vary according to the culture, ethnic group, or socio-economical level of each region [65]. With relation to the children and adolescents, it is important to keep in mind that the consumption choices, especially in children and adolescents that purchase food inside and outside of schools, can be underreported due to stigma or fear of judgement from their parents or principal guardians now of reporting information about consumption [66].

Another limitation to consider is that the present study did not include opportunity costs, that is, the economic value of invested time for the acquiring and preparing of foods. This is relevant in an urban environment where the family’s work and the social dynamics give a different value to time, and it is possible to spend time in certain food-related intra-family activities. It is suggested that future research should include these socioeconomic factors, as well as a more detailed analysis of the nutritional content of the Mexican diet and its cost, and how these decisions can impact the health of the population.

## 5. Conclusions

In conclusion, the dietary patterns of children and adolescents in Mexico City and the Metropolitan Area do not differ in costs from one to another; therefore, the perception that a healthy diet is more expensive can be demystified. More research is needed to investigate the impact of the prices over other Mexican diets in different ages and regions of the country either to reproduce the results of the present study or to learn if differences remain in other regions of the country since the diversity of regional foods in our country is high. Moreover, more research is needed to generate more strategies and public policies that allow Mexican families to learn the components of a healthy diet and how it can be adapted to their monetary capacities.

## Figures and Tables

**Figure 1 nutrients-13-03871-f001:**
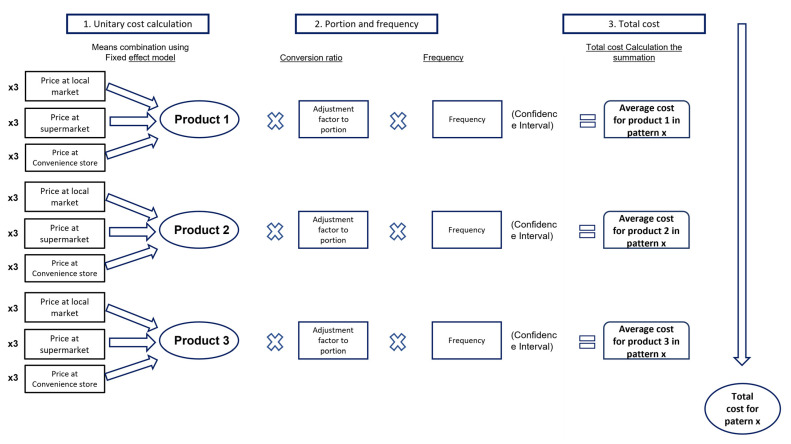
Steps in the cost development of dietary patterns in Mexican children and adolescents; X3 = Acquisition cost sampling of the 146 foods in the FFQ in 3 different stores/market establishments (supermarket, market, and convenience store).

**Figure 2 nutrients-13-03871-f002:**
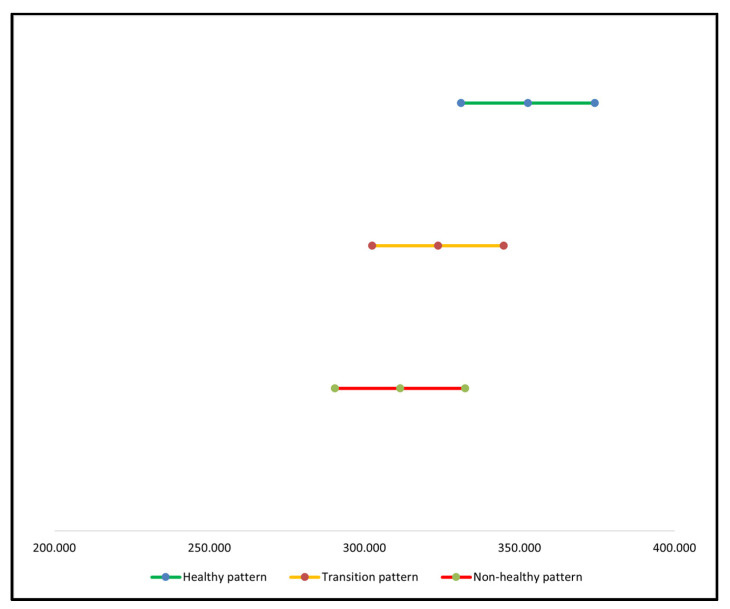
Cost interval of each dietary pattern.

**Table 1 nutrients-13-03871-t001:** Characteristics of the Mexican children and adolescents study sample.

	Boys *n* = 635	Girls *n* = 600	Total *n* = 1235
Age (years)	13.3 (1.4)	12.7 (1.3)	13.0 (1.3)
Anthropometry			
Weight (Kg)	56.8 (17.6)	53.5 (13.1)	55.0 (14.6)
Size (cm)	161.5 (12.9)	155.7 (7.8)	158.4 (10.9)
BMI ^1^ (kg/m^2^)	21.4 (4.3)	21.9 (4.3)	21.7 (4.3)
BMI > p 85 (%)	31.6	31.9	31.8
Waist circumference ^2^ (cm)	75.2 (13.3)	76.8 (12.2)	76.0 (12.7)
CC > p 75 (%)	13.1	44.3	30.1
Body fat percentage	24.7 (10.9)	36.4 (8.8)	31.1 (11.4)
Sexual maturity (%)			
Tanner 1 and 2	34.7	16.1	24.5
Tanner 3 and 4	58.7	70.2	64.9
Tanner 5	6.6	13.7	10.5
Diet			
Total calories (kcal/day) †	2770 (1254)	2318 (1086)	2525 (1186)

^1^ Body mass index: > P85 risk of overweight and/or obesity and ≤ P85 as normal weight. ^2^ Waist circumference: > P75 risk of central obesity and ≤ P75 as normal (according to sex and age) † Information presented in mean values (standard deviation), unless otherwise stated.

**Table 2 nutrients-13-03871-t002:** Example of food and drinks unit cost.

Product	Measurement Unit ^2^	Unit Cost (MXN ^1^)	Conversion of Purchase Unit to Ration	Conversion to the Mexican Equivalents System
Whole milk	1 L	$20.05	240 ml	0.24
Banana	1 Kg	$20.00	80 gr	0.08
Pastries	1 Kg	$55.00	17 gr	0.017
Corn tortilla	1 Kg	$15.00	30gr	0.30
Soft Drinks	1 L	$13.00	89ml	0.089
Plain water	1L	$12.00	240ml	0.24
Egg	1 Kg	$36.70	50 gr	0.05
Pork meat	1 Kg	$80.00	40 gr	0.04
Fritters	1 Kg	$115.00	19 gr	0.019
Lettuce	1 Kg	$42.50	141 gr	0.141

^1^ MXN = Mexican pesos. ^2^ L = liter, Kg = kilogram.

**Table 3 nutrients-13-03871-t003:** Portion consumption within food groups according to dietary patterns.

	Healthy Pattern		Transition Pattern		Non-Healthy Pattern	
Food	Mean	DE	Mean	DE	Mean	DE
Corn and derivatives	3.2	2.5	4.8	2.9	2.9	2.5
Mexican fried food	3.0	2.0	4.6	4.1	4.4	4.0
Wheat and derivatives	4.0	2.8	5.0	3.9	4.5	3.1
Pastries	4.2	3.1	5.9	4.9	3.9	2.9
High fiber cereals	1.1	1.1	0.9	1.2	1.2	1.3
Low fiber cereals	1.1	1.4	1.5	1.6	1.2	1.3
Tuberous root	1.1	0.9	1.2	1.1	1.0	1.0
Rice and pasta	6.1	4.1	7.5	4.8	5.9	3.7
Alcohol	0.2	0.6	0.3	0.8	0.3	0.8
Legumes	3.7	2.7	2.8	2.0	2.9	2.7
Fresh fruits	14.4	6.4	9.0	4.9	9.4	4.8
Fresh vegetables	3.0	1.9	2.3	1.8	2.0	1.6
Industrialized juice	1.3	1.1	1.4	1.0	0.9	0.8
Chicken	3.6	2.4	3.3	1.9	4.4	3.0
Red meat	5.4	2.8	5.2	2.5	7.2	3.8
Processed meat	3.3	1.6	3.9	2.5	5.2	2.8
Fish and seafood	2.6	1.7	1.8	1.3	2.8	1.9
Milk	8.0	5.2	5.9	4.7	7.0	4.9
Dairy products	10.0	5.3	7.6	3.6	11.4	6.1
Egg	0.9	0.6	0.8	0.8	1.3	1.2
Oil seeds and oleaginous fruits	1.9	2.1	1.1	1.4	1.2	1.4
Fats	2.9	1.7	3.4	2.2	4.1	2.5
Candy	5.1	3.2	5.4	3.7	4.2	2.4
Sugar sweetened beverages	4.7	4.0	5.1	4.2	4.2	3.7
Soft Drinks	1.0	1.4	3.3	4.5	2.1	3.6
Fried foods	0.9	0.8	1.6	1.5	1.0	0.9
Other beverages	3.3	2.7	4.3	3.5	3.1	2.3
Purified water	0.1	0.1	0.1	0.1	0.1	0.1
Energy drinks	0.2	0.1	0.3	0.3	0.3	0.3

Information presented in mean values (standard deviation).

**Table 4 nutrients-13-03871-t004:** Mean cost of dietary patterns.

Pattern	Average Cost (MXN ^1^)	Lower Limit (MXN ^1^)	Higher Limit (MXN ^1^)	Average Cost (USD ^2^)	Kcal Energy
Healthy pattern	352.69	331.10	374.27	16.41	2755.8
Transition pattern	323.65	302.46	344.85	15.06	2804.6
Non-healthy pattern	311.43	290.45	332.42	14.49	2813.7

^1^ MXN = Mexican peso. ^2^ Dollar cost calculated for the period starting 1 January until 31 December 2020 (21.496 Mexican pesos to 1 American dollar).

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
