# Peer review of "A Healthy Diet Is Not More Expensive than Less Healthy Options: Cost-Analysis of Different Dietary Patterns in Mexican Children and Adolescents"

_nutrients, 2021, doi:10.3390/nu13113871_

Round 1
Reviewer 1 Report
Comments for Authors
Thank you for the opportunity to review the article entitled: A healthy diet is not more expensive than less healthy options: Cost-analysis of different feeding patterns in Mexican children and adolescents.
The work takes up a very important topic, thank you the authors for taking care of this issue.
I have some minor comments for the authors:
Please revise the keywords to be more specific and use either singular or plural consistently, not alternating e.g. child and adolescents.
In line 92 the authors write about the criteria.
Please add what inclusion and exclusion criteria were used.
Please state the age of the children participating in the study from ... to ...
In tab. 1 we have the average age of the respondents.
The authors divided the group according to gender, I think it is also worth dividing by age groups (the time spent on a computer screen, the amount of sleep, the level of physical activity will be completely different.).
Please add information on how the physical activity of the studied children was tested.
Did the children fill in the questionnaire alone or in the presence of their parents / guardians, does it depend on the age of the child? What criteria were used?
On lines 259-263 the authors use the words "men" and women "and the table shows" boys and girls. "Please use one term consistently.
Please explain all abbreviations used under the tables.
Author Response
Dear Reviewer,
Thank you for giving us the opportunity to submit a revised draft of our manuscript titled A healthy diet is not more expensive than less healthy options: Cost-Analysis of different feeding patterns in Mexican children and adolescents to Nutrients.
Please find attached our response in bold to your comments.
Sincerely
Carlos Mendoza-Gutierrez
Corresponding Author

Reviewer 2 Report
After reading the manuscript I would like to suggest to rethink some issues in some parts of the manuscript.
Introduction
I propose to work through the Introduction. More attention should be paid to the cost of a healthy diet and its assessment. Then it is advisable to present the results of the research conducted so far in other countries and in Mexico - this will justify why the research is important.
Material and Methods
Despite citing another article [23], a short information about the study sample would be appreciated.
Details related to the extraction of factors are needed, i.e. what were the factor loadings, tests used and other necessary indicators, which are usually presented in factor analysis
Line 114. Information about the micro-costing technique is needed, also a reference from the literature
Lines 114-116 Why information about the recruitment period appears here - was the period of collecting information about prices the same as the period of nutritional research among children and adolescents?
Lines 142-150 Usually in the manuscript those questions / variables are described in detail that are used in the analysis - many more variables are indicated here.
Derivation of feeding patterns is rather part of statistical analysis not ‘Measurement instruments’. I suggest to use ‘dietary patterns’ instead of ‘feeding patterns’ in the text.
Lines 165 -167. The sentence “The scores for the factors of each dietary pattern were estimated by adding the consumption of the pondered food groups to the charge of its factors, and each participant received a score for each of the identified patterns” should be rewritten One may have problem with understanding the procedure. .
The Figure 1 should be enlarged – as it stands it is difficult to see what is written
Results
Lines 252-257. These are not the results of the study
Line 259 My suggestion is to use ‘sample’ instead of ‘population’
Table 1. This is not a characteristic of Mexican children, but of the study group - it is necessary to change the title of the table. Some information in the table may be omitted, e.g. for physical activity, while others may be presented in more detail (range) rather than just the average value. After all, a detailed description is presented in the quoted article [23].
Table 2. It would be good to know where the statistically significant differences were.
Lines 297-299. It would be better to use monthly monetary income in the sample to show how much the cost of the healthy pattern and other patterns take in monthly income.
Lines 324-378 This is not a discussion of the results, but a description of the methods used. Brief information on this subject should be included in the Introduction. In the discussion, this information can be used to indicate to what extent the results may be a consequence of the method used. In addition, the results of other studies are described in the discussion, or rather, I would expect an explanation of the obtained results. Did the method of calculating costs, extracting dietary patterns or the socio-economic specificity of Mexico decide about the lack of differences between the cost of individual dietary patterns, or maybe the age of the study group?
The conclusion concerning the belief that a healthy diet is more expensive ‘is demystified’ is too strong. I suggest using word ‘can’ and proposing further research using different age groups, area etc.
Author Response
Dear Reviewer,
Thanks for giving us the opportunity to submit a revised draft of our manuscript titled A healthy diet is not more expensive than less healthy options: Cost-analysis of different feeding patterns in Mexican children and adolescents to Nutrients
Please find attached our response in bold to your valuable comments.
Sincerely
Carlos Mendoza-Gutierrez
Corresponding Author

Round 2
Reviewer 2 Report
I appreciate the changes made by the authors in the manuscript, however I still feel unsatisfied.
I have marked in red bold those remarks that, in my opinion, still need to be rethought and changed or explained why the authors left them in their present form
Introduction
I propose to work through the Introduction. More attention should be paid to the cost of a healthy diet and its assessment. Then it is advisable to present the results of the research conducted so far in other countries and in Mexico - this will justify why the research is important.
Unfortunately I did not find results from other countries – only one reference from 2013 – systematic review. In addition, production costs rather than consumption were taken into account
Material and Methods
Despite citing another article [23], a short information about the study sample would be appreciated.
Details related to the extraction of factors are needed, i.e. what were the factor loadings, tests used and other necessary indicators, which are usually presented in factor analysis.
This information has been included, although I still question whether the method of selecting food patterns should be described in the Measurement instruments section (lines 181-204). This is part of the statistical analysis or results.
Line 114. Information about the micro-costing technique is needed, also a reference from the literature
Lines 114-116 Why information about the recruitment period appears here - was the period of collecting information about prices the same as the period of nutritional research among children and adolescents?
Why have you deleted the information about the period of collecting prices (Lines 289-294). It is important but not in this section.
Lines 142-150 Usually in the manuscript those questions / variables are described in detail that are used in the analysis - many more variables are indicated here.
They are still in the manuscript. Please explain why, for example Table 1, IPAQ – lines 16-179
Derivation of dietary patterns is rather part of statistical analysis not ‘Measurement instruments’.
It has not been changed – please explain the reason.
I suggest to use ‘dietary patterns’ instead of ‘feeding patterns’ in the text.
Lines 165 -167. The sentence “The scores for the factors of each dietary pattern were estimated by adding the consumption of the pondered food groups to the charge of its factors, and each participant received a score for each of the identified patterns” should be rewritten One may have problem with understanding the procedure..
The Figure 1 should be enlarged – as it stands it is difficult to see what is written
Results
Lines 252-257. These are not the results of the study
Line 259 My suggestion is to use ‘sample’ instead of ‘population’
See: the title of Table1
Table 1. This is not a characteristic of Mexican children, but of the study group - it is necessary to change the title of the table. Some information in the table may be omitted, e.g. for physical activity, while others may be presented in more detail (range) rather than just the average value. After all, a detailed description is presented in the quoted article [23].
It was not changed – please explain that
Table 2. It would be good to know where the statistically significant differences were.
Lines 297-299. It would be better to use monthly monetary income in the sample to show how much the cost of the healthy pattern and other patterns take in monthly income.
Lines 324-378 This is not a discussion of the results, but a description of the methods used. Brief information on this subject should be included in the Introduction.
Please explain why this comment was not taken into account
In the discussion, this information can be used to indicate to what extent the results may be a consequence of the method used. In addition, the results of other studies are described in the discussion, or rather, I would expect an explanation of the obtained results. Did the method of calculating costs, extracting dietary patterns or the socio-economic specificity of Mexico decide about the lack of differences between the cost of individual dietary patterns, or maybe the age of the study group?
The discussion was completed with only two paragraphs. However, the discussion as a whole has not changed significantly, so my comments are still valid.
The conclusion concerning the belief that a healthy diet is more expensive ‘is demystified’ is too strong. I suggest using word ‘can’ and proposing further research using different age groups, area etc.
